# The General Linear Cartan Khronon †

**Tomi Koivisto** [1,2,3,*] **, Manuel Hohmann** [2] **and Tom Złośnik** [4]

[1]   Nordita, KTH Royal Institute of Technology and Stockholm University, Roslagstullsbacken 23, SE-10691 Stockholm, Sweden
[2]   Laboratory of Theoretical Physics, Institute of Physics, University of Tartu, W. Ostwaldi 1, 50411 Tartu, Estonia
[3]   National Institute of Chemical Physics and Biophysics, Rävala pst. 10, 10143 Tallinn, Estonia
[4]   CEICO, Institute of Physics of the Czech Academy of Sciences, Na Slovance 1999/2, 182 21 Prague, Czech Republic
*   Correspondence: tomik@astro.uio.no
†   This paper is based on the discussions at the Teleparallel Universes in Salamanca, Salamanca, Spain, 26–28 November 2018.

**Abstract:** A Cartan geometry of the General Linear symmetry is formulated by dividing out the displacements from the group. The resulting action is quadratic in curvature, polynomial in all the (minimal) variables, and describes an observer space that—in the symmetry-broken phase—reproduces the predictions of General Relativity in the presence of dark matter.

**Keywords:** conformal and metric-affine gauge theories of gravitation; Cartan geometry

---

All energy and momentum source gravity. Based on this observation alone, one naturally entertains the idea that "gravity is that field which corresponds to a gauge invariance with respect to displacement transformations" [1]. It is often considered that torsion $T^a$ is the field strength of displacements. However, since the coframe one-form $e^a = t^a + \mathcal{D}x^a$ [2] is the displacement gauge potential $t^a$ only up to the covariant derivative $\mathcal{D} = d + \omega$, where $\omega$ is the Lorenz connection, of a coordinate-scalar $x^a$—the torsion [3]

$$T^a = \mathcal{D}e^a = \mathcal{D}t^a - R_b{}^a x^b \tag{1}$$

is equal to the displacement field strength $T^a = \mathcal{D}t^a$, only if the curvature $R_a{}^b = \mathcal{D}\omega_a{}^b$, vanishes—i.e., only if teleparallelism $R_a{}^b = 0$, is assumed. In this sense, teleparallel gravity [4] can be regarded as a gauge theory of displacements, but so can just as well the more conventional description of gravity without torsion, since if $T^a = 0$, the displacement gauge field strength $\mathcal{D}t^a = R_b{}^a x^b$ is in turn directly proportional to the curvature. In fact, this form is manifested in the recently discovered action that is polynomial in all the variables and quadratic in curvature

$$L = -\frac{1}{2}\left(\frac{1}{8}x^2\epsilon^{ac}{}_{bd} + ix^a x^c \eta_{bd}\right) R_a{}^b \wedge R_c{}^d , \tag{2}$$

but reproduces the predictions of General Relativity in the presence of geometrical dark matter and is possibly capable of extending the description of spacetime into a pregeometric phase, without invoking any (co)frame field [5] (the four-form (2) is considered as a function of only $\omega_a{}^b$ and $x^a$, with $\eta_{ab}$ and $\epsilon_{abcd}$ being the two invariants of the special orthogonal algebra). These are fresh insights, but follow closely Cartan's original developments [3], where attention was directed to the case wherein the displacement gauge potential vanishes, $t^a = 0$. In such a case, the field $x^a$ is known as the Cartan

radius vector [6]. In the context of the new theory [5], we might call it the *Cartan Khronon* since it is plays the role of a symmetry-breaking field that determines the direction of time (which only exists in the geometric, symmetry-broken phase).

Besides a Cartan-geometric, a canonical gauge theory of displacements [7] has also been taken into reconsideration in the current literature [8–11]. A displacement transformation can be seen, from the passive perspective, as a general coordinate transformation. Obviously, such a transformation is included in the GL (General Linear) transformations[1]. In a GL gauge connection, the displacement component is precisely that which has neither curvature nor torsion[2] [8]. The reformulation of Einstein's theory in such a geometrical setting, which is "purified" from all non-integrable elements, allows the localization of the gravitational energy and the path integral formalism that does not suffer from the necessity of boundary terms [12]. The unique property of the gravitational interaction, that it can be always locally eliminated, is reflected in the integrability of the connection [13].

Thus, both the canonical and the Cartan-geometric approaches to gauge theory of displacements suggest that $\mathcal{D}t^a = 0$, and the physical interpretation of this is the essence of General Relativity, the equivalence of gravity and inertia. In this note, we shall find some further connections between purified gravity and Cartan geometry [3,14–17]. The purpose of this note is to formulate a gauge theory of displacements in terms of a GL-covariant (exterior) derivative D, and two symmetry-breaking fields **x** and **y**. The spacetime metric will then be an emergent structure that only exists in the symmetry-broken phase.

In the conventional metric-affine gauge theories of gravitation [6], besides invoking the metric as an additional field, one gauges both the GL symmetry and displacements. In the language of Cartan geometry, the model is $G/H$, where $G = \mathrm{GA}(4, \mathbb{R}) = \mathrm{GL}(4, \mathbb{R}) \ltimes \mathbb{R}^4$, sometimes called the general affine group, is an inhomogenisation of the $H = \mathrm{GL}(4, \mathbb{R})$. However, this is superfluous since the displacements are already included in the GL as pointed out above. Actually, there are two (coupled) sets of displacements. This can be made apparent in the spinor representation in terms of the Dirac matrices $\gamma_a$, which satisfy the Clifford algebra $\{\gamma_a, \gamma_b\} = -2\eta_{ab}$. We have the left and the right displacements, generated by

$$\overset{+}{\gamma}_a = \frac{1}{2}\left(1 + \gamma_5\right)\gamma_a, \quad \overset{-}{\gamma}{}^a = \frac{1}{2}\left(1 - \gamma_5\right)\gamma^a, \tag{3}$$

which form two Abelian subgroups. We write the full GL connection and its curvature as

$$\mathrm{D} = \mathrm{d} + t^a\overset{+}{\gamma}_a + s_a\overset{-}{\gamma}{}^a + \frac{1}{2}\omega_a{}^b i\sigma^a{}_b + p\frac{-\gamma_5}{2} + q, \quad \mathbf{F} = \tilde{T}^a\overset{+}{\gamma}_a + \tilde{S}_a\overset{-}{\gamma}{}^a + \frac{1}{2}R_a{}^b i\sigma^a{}_b - \frac{1}{2}P\gamma_5 + Q, \tag{4}$$

respectively. Here, $\sigma_{ab} = i\gamma_{[a}\gamma_{b]}$ and $s^a$, $p$, $q$ are further components of the gauge potential. On this basis, we may recognize the connection of conformal gauge theory [18] plus the trivial gauge field $q$. Explicitly, the field strength components in (4) are

$$\tilde{T}^a = \mathrm{d}t^a - \omega_b{}^a \wedge t^b - p \wedge t^a, \tag{5a}$$

$$\tilde{S}_a = \mathrm{d}s_a + \omega_a{}^b \wedge s_b + p \wedge s_a, \tag{5b}$$

$$R_a{}^b = \mathrm{d}\omega_a{}^b - \omega_a{}^c \wedge \omega_c{}^b - 2t^{[b} \wedge s_{a]}, \tag{5c}$$

$$P = \mathrm{d}p - t^a \wedge s_a, \tag{5d}$$

$$Q = \mathrm{d}q. \tag{5e}$$

---

[1]　This is contrary to what was implied in references [10,11]. We agree with reference [11] that the principal bundle of the displacement group is not a vector bundle, since in gauge theories the fibers are actually torsors instead of groups (which does not change the conclusion of references [10] that such a displacement bundle is topologically trivial).

[2]　Consider an affine connection $\Gamma^\alpha{}_{\mu\nu}$ in the tensor representation. If without curvature, it will be given by a GL transformation $\Lambda^\alpha{}_\beta$ of the zero connection as $\Gamma^\alpha{}_{\mu\nu} = \Lambda^\alpha{}_\beta \partial_\mu (\Lambda^{-1})^\beta{}_\nu$, and if one further assumes the vanishing of torsion $T^\alpha{}_{\mu\nu} = 2\Gamma^\alpha{}_{[\mu\nu]} = 0$, it follows further that $\Lambda^\alpha{}_\beta = \partial_\beta \xi^\alpha$, which is nothing but the Jacobian of a coordinate transformation.

One can straightforwardly check that the above-specified 16 matrices generate the GL. The nonvanishing algebra includes the Poincaré subalgebras

$$[\sigma_{ab}, \sigma_{cd}] = 2i\left(\eta_{c[a}\sigma_{b]d} - \eta_{d[a}\sigma_{b]c}\right), \quad [\overset{\pm}{\gamma}_a, \sigma_{bc}] = 2i\eta_{a[c}\overset{\pm}{\gamma}_{b]}, \tag{6}$$

and

$$[\overset{+}{\gamma}_a, \overset{-}{\gamma}_b] = 2\left(\eta_{ab}\frac{-\gamma_5}{2} - i\sigma_{ab}\right), \quad [\overset{\pm}{\gamma}_a, \frac{-\gamma_5}{2}] = \pm\overset{\pm}{\gamma}_a, \tag{7}$$

is the rest of the algebra. Note in particular the presence of an 8-dimensional subgroup $H$ generated by $\sigma^a{}_b, \gamma_5$ and the central element 1 of the Clifford algebra, which together with $G = $ GL may define a Cartan gauge theory modeled on $G/H$. More conventional 4-dimensional model geometry is achieved instead by also including one set of displacements into the $H$ that is then 12-dimensional. From the algebra relations we can determine the Cartan–Killing form[3]. Its nonzero components dictate that

$$\langle \sigma^{ab}\sigma_{cd}\rangle = 4\delta_c^{[a}\delta_d^{b]}, \quad \langle \overset{+}{\gamma}_a\overset{-}{\gamma}_b\rangle = \langle \overset{-}{\gamma}_a\overset{+}{\gamma}_b\rangle = -2\eta_{ab}, \quad \langle \gamma_5\gamma_5\rangle = 4. \tag{8}$$

Thus, we have a group-invariant definition of an inner product.

In quantum mechanics, complex structure is not optional. In the present context of GL gauge theory, it is necessary, for the purpose of incorporating spinors, to consider the group GL(4,$\mathbb{C}$) over complex numbers, since the GL(4,$\mathbb{R}$) over real numbers does not have finite-dimensional spinors [6]. It is convenient then to take $\eta_{ab} = -\delta_{ab}$ (and thus, Euclidean gamma matrices). Though GL(4,$\mathbb{C}$) has the real dimension 32, we would like to point out in passing that actually, there may nothing extra in the connection (4) if one is eventually aiming at a unification: Like in Plebanski's and Ashtekar's gravity [19], the spin connection is the self-dual part of $\omega_a{}^b$; like in the Cartan Khronon theory [5], the real component of the anti-self dual part of $\omega_a{}^b$ could be essentially the triad; as in the graviweak theory, [20] the imaginary component of the anti-self dual could be the SU$_L$(2); and along the lines of Weyl's gauge theories [21,22], the real part of the dilaton $p$ could gauge the scale invariance and the imaginary part the U$_Y$(1). Since in the spinorial representation (6,7) it becomes apparent that the SL(4,$\mathbb{C}$) double-covers the SO(6,$\mathbb{C}$), we have dubbed this the Conformal Affine Theory. An interesting property is that the $-\gamma_5/2$ has the role of the generator of dilatons in spacetime geometry. This hints at the connection between spatial length scales and particle mass scales, since in quantum field theory the appearance of mass terms for matter fields occur via left–right couplings. In this note, however, our focus is on gravity and the displacements (3).

We now begin the Cartan construction with two spacetime scalars $\mathbf{x} = x^a\overset{+}{\gamma}_a$ and $\mathbf{y} = y_a\overset{-}{\gamma}^a$,

$$\mathbf{e} = \mathbf{t} + \mathrm{D}\mathbf{x} = \left[t^a + (\mathrm{d} - p)\,x^a - \omega_b{}^a x^b\right]\overset{+}{\gamma}_a + s_a x^a \gamma_5 + 2s_a x^b i\sigma^a{}_b, \tag{9a}$$

$$\mathbf{\partial} = \mathbf{s} + \mathrm{D}\mathbf{y} = \left[s_a + (\mathrm{d} + p)\,y_a + \omega_a{}^b y_b\right]\overset{-}{\gamma}^a - t^a y_a \gamma_5 + 2t^b y_a i\sigma^a{}_b. \tag{9b}$$

Only in the gauge $\mathbf{t} = 0$, the $\partial$ becomes a conventional frame field for any $\mathbf{y}$; and only in the gauge $\mathbf{s} = 0$, the $\mathbf{e}$ becomes a conventional coframe field for any $\mathbf{x}$. The availability of these gauge choices requires that $\tilde{\mathbf{T}} = 0$ and $\tilde{\mathbf{S}} = 0$, respectively. The torsion and cotorsion are

$$\mathbf{T} = \mathrm{D}\mathbf{e} = \left(\tilde{T}^a - R_b{}^a x^b - Px^a\right)\overset{+}{\gamma}_a + \left(s_a \wedge t^a + \tilde{S}_a x^a\right)\gamma_5 - 2\left(s_a \wedge t^b + \tilde{S}_a x^b\right)i\sigma^a{}_b, \tag{10a}$$

$$\mathbf{S} = \mathrm{D}\mathbf{\partial} = \left(\tilde{S}_a + R_a{}^b y_b + Py_a\right)\overset{-}{\gamma}^a - \left(s_a \wedge t^a + \tilde{T}^a y_a\right)\gamma_5 + 2\left(s_a \wedge t^b + \tilde{T}^b y_a\right)i\sigma^a{}_b. \tag{10b}$$

---

[3]　Or alternatively, from the traces of the matrix products, noting that for GL(n) matrices $\mathbf{M}_1$ and $\mathbf{M}_2$ we have $\langle\mathbf{M}_1\mathbf{M}_2\rangle = \mathrm{Tr}(\mathbf{M}_1\mathbf{M}_2) - \mathrm{Tr}(\mathbf{M}_1)\mathrm{Tr}(\mathbf{M}_2)/n$. To display the algebra compactly (6–8), we have raised and lowered the indices of the generators with the metric $\eta_{ab}$, though elsewhere in this note we avoid that.

Consider the gauge transformation

$$\epsilon = \overset{+}{\epsilon}{}^{a}\overset{+}{\gamma}_{a} + \overline{\epsilon}_{a}\overline{\gamma}{}^{a} + \frac{1}{2}\epsilon_{a}{}^{b}i\sigma^{a}{}_{b} - \frac{1}{2}\epsilon\gamma_{5} + \varepsilon, \tag{11}$$

where $\eta^{ca}\epsilon_{c}{}^{b} = \eta^{c[a}\epsilon_{c}{}^{b]}$. The gauge potentials transform as

$$\delta_{\epsilon}t^{a} = -(\mathrm{d} - p)\overset{+}{\epsilon}{}^{a} + \omega_{b}{}^{a}\overset{+}{\epsilon}{}^{b} - t^{b}\epsilon_{b}{}^{a} - t^{a}\epsilon, \tag{12a}$$

$$\delta_{\epsilon}s_{a} = -(\mathrm{d} + p)\overline{\epsilon}_{a} - \omega_{a}{}^{b}\overline{\epsilon}_{b} + s_{b}\epsilon_{a}{}^{b} + s_{a}\epsilon, \tag{12b}$$

$$\delta_{\epsilon}\omega_{a}{}^{b} = -\mathrm{d}\epsilon_{a}{}^{b} - 2\omega_{c}{}^{[a}\epsilon_{b]}{}^{c} - 4\left(s_{[a}\overset{+}{\epsilon}{}^{b]} - t^{[a}\overline{\epsilon}_{b]}\right), \tag{12c}$$

$$\delta_{\epsilon}p = -\mathrm{d}\epsilon - 2\left(s_{a}\overset{+}{\epsilon}{}^{a} - t^{a}\overline{\epsilon}_{a}\right), \tag{12d}$$

$$\delta_{\epsilon}q = -\mathrm{d}\varepsilon, \tag{12e}$$

and the two GL-charged coordinate-scalars transform as

$$\delta_{\epsilon}\mathbf{x} = \left(\overset{+}{\epsilon}{}^{a} - \epsilon_{b}{}^{a}x^{b} - \epsilon x^{a}\right)\overset{+}{\gamma}_{a} + \overline{\epsilon}_{a}x^{a}\gamma_{5} + 2\overline{\epsilon}_{a}x^{b}i\sigma^{a}{}_{b}, \tag{13a}$$

$$\delta_{\epsilon}\mathbf{y} = \left(\overline{\epsilon}_{a} + \epsilon_{a}{}^{b}y_{b} + \epsilon y_{a}\right)\overline{\gamma}{}^{a} - \overset{+}{\epsilon}{}^{a}y_{a}\gamma_{5} + 2\overset{+}{\epsilon}{}^{a}y_{b}i\sigma^{b}{}_{a}. \tag{13b}$$

We can check the transformations of the components $\mathrm{e}^{a}$ and $\partial_{a}$ to verify:

- If we fix $\overline{\epsilon}$ such that $\mathbf{s} = 0$, the cotetrad transforms as $\delta_{\epsilon}\mathrm{e}^{a} = -\epsilon_{b}{}^{a}\mathrm{e}^{b} + \epsilon\mathrm{e}^{a}$, in particular, $\delta_{\overset{+}{\epsilon}}\mathbf{e} = 0$.

- If we fix $\overset{+}{\epsilon}$ such that $\mathbf{t} = 0$, the tetrad transforms as $\delta_{\epsilon}\partial_{a} = \epsilon_{a}{}^{b}\partial_{b} - \epsilon\partial_{a}$, in particular, $\delta_{\overline{\epsilon}}\partial = 0$.

Imposing both would essentially leave us with the centrally extended Weyl group (generated by the central element 1, $\sigma^{a}{}_{b}$, and $\gamma_{5}$) properly gauged, and imposing one of the above conditions would leave us with the centrally extended inhomogeneous Weyl group (which has either the $\overline{\gamma}{}^{a}$ or the $\overset{+}{\gamma}_{a}$ as additional generators) properly gauged. It is because the Cartan geometry is not reductive [14], that we need to impose some restrictions on the solutions.

In either context[4] we can embed the Cartan Khronon into the GL gauge theory. Using the Cartan–Killing form (8), one can write an invariant Lagrangian four-form involving the GL curvature and the Cartan Khronon as follows:

$$L = -\frac{i}{8}\left(\langle \mathbf{x}\mathbf{F} \wedge \mathbf{y}\mathbf{F}\rangle + \frac{1}{2}\langle [\mathbf{x}, \mathbf{y}]\mathbf{F} \wedge \mathbf{F}\rangle - \frac{1}{4}\langle \mathbf{x}\mathbf{y}\rangle\langle \mathbf{F} \wedge \mathbf{F}\rangle\right). \tag{14}$$

We have chosen the coefficients of the invariants such that by setting $\mathbf{F} \to \mathbf{R}$ and $y_{a} \to \eta_{ab}x^{b}$ (14) reduces precisely to[5] (2). We have not introduced a metric by hand, but its components $g_{\mu\nu}$ will correspond to $-\frac{1}{2}\langle \partial, \mathbf{e}\rangle \to \eta_{ab}\mathcal{D}x^{a} \otimes \mathcal{D}x^{b} = g_{\mu\nu}\mathrm{d}x^{\mu} \otimes \mathrm{d}x^{\nu}$. The compatible connection is given by $-2\Gamma^{\alpha}{}_{\mu\nu} = \partial_{a}{}^{\alpha}\mathcal{D}_{\mu}\mathrm{e}^{a}{}_{\nu} = -\mathrm{e}^{a}{}_{\nu}\mathcal{D}_{\mu}\partial_{a}{}^{\alpha}$, whereas the connection of purified gravity is simply given by the same expression when $\mathcal{D} \to \mathrm{d}$.

Finally, we note that the GL Cartan Khronon can be considered as the single field $\mathbf{z} = (\mathbf{x} + i\mathbf{y})/2$. This allows the replacement of (14) by

$$L = \frac{1}{4}\left(\langle \mathbf{z}\mathbf{z}^{*}\mathbf{F} \wedge \mathbf{F}\rangle - \langle \mathbf{z}\mathbf{F} \wedge \mathbf{z}\mathbf{F}\rangle + \frac{1}{4}\langle \mathbf{z}\mathbf{z}\rangle\langle \mathbf{F} \wedge \mathbf{F}\rangle\right). \tag{15}$$

---

[4]  There would be interesting alternative ways to proceed [13,23], but we need not be concerned with them in this note.
[5]  It seems evident that the equations of motion for $s_{a}$ and $t^{a}$ are trivially satisfied by $s_{a} = t^{a} = 0$. Note that the dilation invariance is retained.

If we also define $æ = \frac{1}{2}(\mathbf{e} + i\mathbf{ə}) = \frac{1}{2}(\mathbf{t} + i\mathbf{s}) + D\mathbf{z}$, the spacetime metric and connection will then correspond to the GL-invariant objects $i\langle æ \otimes æ \rangle \rightarrow g_{\mu\nu}\mathrm{d}x^{\mu} \otimes \mathrm{d}x^{\nu}$ and $i\langle æ \otimes D \otimes æ \rangle \rightarrow \Gamma^{\alpha}{}_{\mu\nu}\partial_{\alpha} \otimes \mathrm{d}x^{\mu} \otimes \mathrm{d}x^{\nu}$, respectively, in the abovementioned limit wherein (15) reduces to the Lorentz gauge theory (2).

To conclude, we have formulated a Cartan-geometric, polynomial GL gauge theory wherein displacements are generated by an Abelian subgroup of the GL. The result, Equation (15), extends General Relativity by eliminating the need for dark matter[6] and by incorporating a symmetric, pregeometric phase. Thus, we can combine the gratifying properties of a canonical displacement gauge theory and the realization of an observer space á la Cartan Khronon. It remains to be seen how far we may proceed with unification from this basis. The challenges would be to assign the gauge fields of the particle interactions within the GL connection and to join the Cartan Khronon and the Higgs that was observed in the Large Hadron Collider into a single scalar. It could also be interesting to consider whether the GL gauge theory could address the cosmological problems of dark energy and the (inflationary) beginning of the universe.

**Author Contributions:** Writing—original draft preparation, T.K.; writing—review and editing, M.H., T.Z.; conceptualization, methodology, validation, formal analysis, investigation, resources, visualization, funding acquisition, M.H., T.K., T.Z.

**Funding:** The work was supported by the Estonian Research Council through the Personal Research Funding project PRG356 "Gauge Gravity" and by the European Regional Development Fund through the Center of Excellence TK133 "The Dark Side of the Universe". TZ is funded by the European Research Council under the European Union's Seventh Framework Programme (FP7/2007-2013)/ERC Grant Agreement n. 617656 "Theories and Models of the Dark Sector: DM, Dark Energy and Gravity".

**Acknowledgments:** This note has benefitted from interesting discussions with Luca Marzola and many of the participants of the workshop *Teleparallel Universes in Salamanca*.

**Conflicts of Interest:** The authors declare no conflict of interest.

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
