# Peer review of "The General Linear Cartan Khronon†"

_universe, doi:10.3390/universe5070168_

Round 1
Reviewer 1 Report
In this paper, the authors summarize the formulation of a Cartan-geometric polynomial GL gauge theory that may have interesting implications.
I understand that this paper reports the contribution to a conference. However, to make it self-contained, I would appreciate if additional explanations were included to clarify the mathematical steps.
In the first place, I recommend to the authors including some explanation of what is done from equation (6) to (8) before they proceed to summarize the maths. What is postulated and what is obtained? What is used to obtain it?
In the second place, in my opinion the paragraph after equation (8), that is “To incorporate spinors…”, contains too many information in few lines. It may be clearer if it is split to include more context and explanation of the different issues.
In the third place, in my opinion the emergence of a metric is a crucial characteristic of their study. So, I think that the impact of the paper may be amplified if they could include further clarifications about this emergence in the paragraph after equation (14) and/or elsewhere.
On the other hand, the authors do not explicitly specify the meaning of some symbols they use when they first appear in the equations. In particular, they should include the definition of epsilon and eta after equation (2) and that of s, sigma, p and q after equation (4).
In summary, I consider that this paper should be published after the authors include clarifications about the issues mentioned above. This would improve the potential impact of the paper.
Author Response
We thank the referee for the careful reading of the manuscript and for the concrete suggestions to improve the paper by including explanations at some points which had been left unclear. Below we specify the modifications that have been implemented at the points that were raised.
"In the first place, I recommend to the authors including some explanation of what is done from equation (6) to (8) before they proceed to summarize the maths. What is postulated and what is obtained? What is used to obtain it?"
The only assumption is GL symmetry: equations (6-8) are properties of the GL algebra. What is nontrivial is that the 16 generators appearing in (4) generate precisely this algebra. Just mentioning that "In this basis we may recognise the connection of conformal gauge theory [18] plus the trivial gauge field q" indeed may leave this unclear! Therefore we have have added the sentence: "One can straightforwardly check that the matrices specified above indeed generate the GL." Thus (6,7) only verify that statement. Then, as further explained in the footnote 3, the Cartan-Killing form (8) is contained in the structure of the algebra."
In the second place, in my opinion the paragraph after equation (8), that is “To incorporate spinors…”, contains too many information in few lines. It may be clearer if it is split to include more context and explanation of the different issues."
We agree with this. We have begun the paragraph now with two more readable sentences, as follows: "In quantum mechanics, complex structure is not optional. In the present context of GL gauge theory, it is necessary for the purpose of incorporating spinors, to consider the group GL(4,C) over complex numbers, since the GL(4,C) over real numbers does not have finite-dimensional spinors [6]." Thus, we also mentioned a reference to the classic reference on the GA(4,R) theory review (which includes a section on the infinite-dimensional objects called world spinors).
p.p1 {margin: 0.0px 0.0px 0.0px 0.0px; font: 11.0px Menlo; background-color: #edfdd5} span.s1 {font-variant-ligatures: no-common-ligatures}"
In the third place, in my opinion the emergence of a metric is a crucial characteristic of their study. So, I think that the impact of the paper may be amplified if they could include further clarifications about this emergence in the paragraph after equation (14) and/or elsewhere."
Since the elimination of the metric as an input of the theory is highlighted at a few points and we can only refer to [5] and a few works on observer space for the detailed implications of such theories as far as they're understood this far, we have only added the final footnote 6 to emphasise another advantageous aspect of the scenario: in particular, in the context of gravitational dark matter alternatives it is common to introduce new fields besides the metric/tetrad, but our construction is in a sense rather a reduction than extension of the standard gauge formulations of the General Relativity theory."
On the other hand, the authors do not explicitly specify the meaning of some symbols they use when they first appear in the equations. In particular, they should include the definition of epsilon and eta after equation (2) and that of s, sigma, p and q after equation (4)."
Done.
Reviewer 2 Report
This communication presents some new results and ideas that can be published. The connection of the results of this paper with the dark matter (which is mention in the abstract) phenomenon is not understandable and may be a comment or an explanation in this direction could be of use for the reader. In this respect the authors may be interested to know that other models that also involve four scalars (to build a non metric measure of integration in the action) have been used to address the appearence of Dark Matter , see for example
A two measure model of dark energy and dark matter |
and many others. In conclusion the paper is publishable but the authors should explain some more about their dark matter ideas in the text of the paper, since they are mentioning the subject in the abstract, the discussion must include citing references like the one I pointed out above.
Author Response
p.p1 {margin: 0.0px 0.0px 0.0px 0.0px; font: 11.0px Menlo; background-color: #edfdd5} span.s1 {font-variant-ligatures: no-common-ligatures}
We thank the referee for the comments. We have added a comment on other gravitational alternatives to dark matter, expanding the reference list with four entries, including the one suggested by the referee. In particular, we mention the interesting non-metric measure, which indeed introduces new possibilities.